# Promptable Closed-loop Traffic Simulation

**Shuhan Tan**[1*]  **Boris Ivanovic**[2]  **Yuxiao Chen**[2]  **Boyi Li**[2]
**Xinshuo Weng**[2]  **Yulong Cao**[2]  **Philipp Krähenbühl**[1]  **Marco Pavone**[2]
[1]UT Austin    [2]NVIDIA

**Abstract:** Simulation stands as a cornerstone for safe and efficient autonomous driving development. At its core a simulation system ought to produce realistic, reactive, and controllable traffic patterns. In this paper, we propose `ProSim`, a multimodal promptable closed-loop traffic simulation framework. `ProSim` allows the user to give a complex set of numerical, categorical or textual prompts to instruct each agent's behavior and intention. `ProSim` then rolls out a traffic scenario in a closed-loop manner, modeling each agent's interaction with other traffic participants. Our experiments show that `ProSim` achieves high prompt controllability given different user prompts, while reaching competitive performance on the Waymo Sim Agents Challenge when no prompt is given. To support research on promptable traffic simulation, we create `ProSim-Instruct-520k`, a multimodal prompt-scenario paired driving dataset with over 10M text prompts for over 520k real-world driving scenarios. Code, data and labeling tools avaliable at https://ariostgx.github.io/ProSim.

**Keywords:** Autonomous Driving, Scenario Generation, Traffic Simulation

## 1 Introduction

Simulation provides an efficient way to evaluate AV systems in realistic environments. It avoids the high costs and risks associated with real-world testing, and forms the first step of any deployment cycle of an autonomous driving policy. A simulator is only able to fulfill its role if it is capable of mimicking real-world driving conditions. Chief among these conditions is highly-realistic simulation of *interactive* traffic agents and their behavior patterns. Further, the simulation must be *controllable* to enable the creation of interesting traffic scenarios by editing and customizing each agent's behaviors and intentions via user-friendly inputs or prompts.

To capture these needs, we propose a new task: *promptable closed-loop traffic simulation* (Section 3). Aside from simulating realistic traffic agent interactions in *closed-loop*, traffic models should also generate agent motions that satisfy a complex set of *user-specified prompts*. Our baseline algorithm `ProSim` (Figure 1) takes a scene initialization and multimodal prompts as inputs, and produces a set of agent policies that yield closed-loop, reactive scenario rollouts. As we show in Section 6, `ProSim` achieves high realism and controllability with a variety complex user prompts. `ProSim` even achieves competitive performance on the Waymo Open Sim Agent Challenge [1] without prompts.

To further enable research into promptable closed-loop traffic simulation task, we created `ProSim-Instruct-520k`, a multimodal paired prompt-scenario dataset (Section 5). Leveraging the real-world AV dataset [2], we provide a rich set of prompts for all agents. Specifically, for each scenario we label agents' Goal Points (their destinations), Action Tag (e.g., *Accelerate*, *LeftTurn*), Route Sketch (a noisy sketch of an agent's route), and Text instructions (e.g., *"Instruct <A0> to decelerate before turning right."*), visualized in Figure 1. These prompts simulate some of the diverse ways users may wish to prompt scenarios. We ensure the labeled prompts accurately depict the agent's ground-truth (GT) behavior with careful human quality assurance stages. `ProSim-Instruct-520k`

---

*Work done during an internship at NVIDIA.

8th Conference on Robot Learning (CoRL 2024), Munich, Germany.

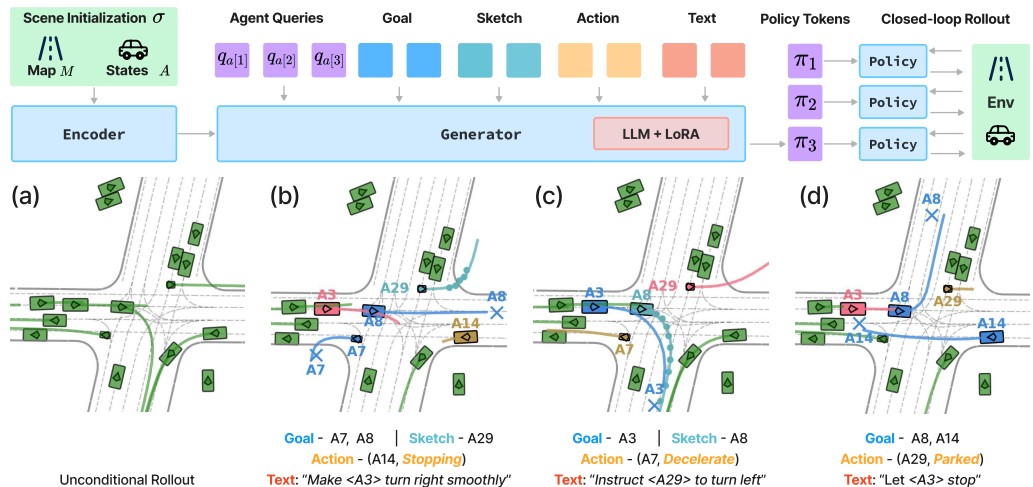

Figure 1: Overview of `ProSim` and promptable closed-loop traffic simulation. All agents are controlled by `ProSim`: green ones are unconditioned and others are prompted with multimodal prompts.

contains 520K prompt-scenario pairs of real-world driving data, including over 10M sentences of text prompts for 10M unique agents and 575 hours of driving. We will release the data, a benchmark, and labeling tools of `ProSim-Instruct-520k` to facilitate future research upon publication.

Our key contributions are threefold:

1. We introduce `ProSim`, a first-of-its-kind promptable closed-loop traffic simulation framework.
2. We create `ProSim-Instruct-520k`, a large-scale multimodal prompt-scenario driving dataset. It is the first driving dataset with semantic-rich agent motion labels and text captions.
3. We will release code and checkpoints of `ProSim`. We will also release data, a benchmark, and the labeling tools of `ProSim-Instruct-520k` to facilitate agent motion simulation research.

## 2 Related Work

**Unconditional closed-loop traffic simulation.** Recent advancements in generative models for closed-loop traffic simulation have garnered significant attention. These works aim to learn realistic multi-agent interactions from real-world data. Early work TrafficSim [3] learns multi-agent interactions with GNNs. More recent works like BITS [4] and TrafficBots [5] learn agent behaviors with a bi-level imitation learning design, while RTR [6] combines reinforcement learning and imitation learning objectives. Most recently, works like Trajeglish [7] and SMART [8] showed the effectiveness of formulating the task as a next-token prediction problem. For this research direction, Waymo Open Sim Agent Challenge [1] provides a common evaluation methodology and benchmark. The objective of these works is to replicate real-world agent interactions without much control over the simulation process. In contrast, `ProSim` focuses on providing users with different ways to *prompt* the agents, allowing efficient creation of interesting traffic scenarios.

**Controllable open-loop traffic generation.** Another line of work focuses on generating traffic scenarios with user-input controls. Early works like CTG [9] and MotionDiffuser [10] generate agent trajectories with diffusion models that imposes control signals with cost functions. CTG++[11] replaces the gradient guidance process in CTG with cost functions generated by an LLM, supporting text conditions as input. Different from these works that assume existing agent locations as input, SceneGen [12] and TrafficGen [13] automatically initialize agent positions conditioned on an empty map. LCTGen [14] further supports both agent placement and motion generation with text conditions as input. RealGen [15] incorporates retrievals of examples to generate desired scenarios given the query. While these approaches can generate realistic scenarios controlled by user inputs, they only produce fixed-length agent trajectories in *open loop*. These trajectory outputs do not allow agents to react to each other or the tested driving policy in a simulator, limiting their usefulness in

closed-loop simulation. In contrast, `ProSim` is explicitly designed for *closed-loop* traffic simulation, yielding natural agent interactions and complex prompt adherence.

## 3  Problem Formulation

In the task of traffic modeling, we are given an initial traffic scene $\sigma = (M, A)$ comprised of map elements $M = \{v_1, ..., v_S\}$ and the initial states of $N$ agents $A = \{a_1, ..., a_N\}$ (i.e., their sizes, types, and past trajectory). Each map element $v$ is a polyline segmentation of a lane into its centerline and edges. Agent states $\mathbf{s}_t = \{s_t^1, ..., s_t^N\}$ are agent positions and headings at time $t$.

To simulate a traffic scenario for $T$ steps, the task of traffic modeling is to obtain the distribution $p(\mathbf{s}_{1:T}|\sigma)$. We denote rollout $\tau \sim p(\mathbf{s}_{1:T}|\sigma)$ as a sample from this distribution. Interactive traffic modeling [1] factorizes the rollout distribution autoregressively, $p(\mathbf{s}_{1:T}|\sigma) = \prod_{t=1}^{T} p(\mathbf{s}_t|\mathbf{s}_{1:t-1}, \sigma)$, independently for each agent, $p(\mathbf{s}_t|\mathbf{s}_{1:t-1}, \sigma) = \prod_{i=1}^{N} p(s_t^i|\mathbf{s}_{1:t-1}, \sigma)$. Thus, the task of *unconditional traffic simulation* [1] reduces to predicting $p(\mathbf{s}_{1:T}|\sigma) = \prod_{t=1}^{T} \prod_{i=1}^{N} p(s_t^i|\mathbf{s}_{1:t-1}, \sigma)$.

In this work, we aim to incorporate user-specified prompts $\rho = \{\rho_1, ..., \rho_N, L\}$, which is a set of prompts for each agent in the scenario as well as an *optional* natural language prompt $L$. For agent $i$, the prompt $\rho_i$ contains its static properties $a_{s[i]}$ and *optionally* a set of user prompts $\{(x_1, \omega_1), ..., (x_{|\rho_i|}, \omega_{|\rho_i|})\}$, where $x$ is the prompt input and $\omega$ is the prompt type (e.g., goal point, action). The text prompt $L$ can include multiple sentences or be empty. Our goal is to generate scenarios following user prompts, yielding the **promptable closed-loop traffic simulation** task:

$$p(\mathbf{s}_{1:T}|\sigma, \rho) = \prod_{t=1}^{T} \prod_{i=1}^{N} p(s_t^i|\mathbf{s}_{1:t-1}, \sigma, \rho). \tag{1}$$

When $T$ is large, modeling the above equation becomes computationally hard. We thus discretize $T$ into smaller $k$-step chunks, and factorize the probability accordingly: $p(\mathbf{s}_{1:T}|\sigma, \rho) = \prod_{t=1}^{T'} \prod_{i=1}^{N} p(\mathbf{s}_{t\cdot k:t\cdot k+k}^i|\mathbf{s}_{1:t\cdot k-1}, \sigma, \rho)$, where $T' = \frac{T-k}{k}$. In this way, we balance computation cost and simulation accuracy by selecting different values for $k$.

## 4  `ProSim`

`ProSim`$(\tau|\sigma, \rho)$ is a prompt-conditioned model that produces interactive traffic rollouts $\tau$ from a scene initialization $\sigma$ and a set of user prompts $\rho$. `ProSim` has three main components: first, a scene `Encoder` (Section 4.1) efficiently encodes an scene initialization $\sigma$ to a shared set of scene tokens $F = $ `Encoder`$(\sigma)$. Secondly, a policy token `Generator` (Section 4.2) takes the scene tokens and agent prompts to generate policy tokens for all agents $\{\pi_1, ..., \pi_N\} = $ `Generator`$(F, \rho)$. Thirdly, for agent $i$ at timestep $t$, a `Policy` network (Section 4.3) takes the policy token $\pi_i$, agent states $\mathbf{s}_{1:t-1}$, and current observation $F$ to predict the next $k$ agent states $\mathbf{s}_{t:t+k}^i = $ `Policy`$(\pi_i, \mathbf{s}_{1:t-1}, F)$. Finally, we obtain the sample rollout $\hat{\tau} \sim $ `ProSim`$(\sigma, \rho)$ by iterating between sampling $\mathbf{s}_{t:t+k}^i$ from `Policy` for each agent and updating the observation $\mathbf{s}_t$ from $t = 1$ to $T$ with $k$-step strides. We train `ProSim` with efficient closed-loop training (Section 4.4) on `ProSim-Instruct-520k`.

`ProSim` prioritizes three key properties. First, it is highly *controllable*. `ProSim` allows users to give a flexible combination of numerical, categorical, and textual prompts for each agent, or no prompt for unconditional driving. Users may also mix and match different prompting strategies. Second, `ProSim` allows interactive *closed-loop* traffic simulation. For each agent, the `Generator` outputs a policy token instead of the full trajectory. This allows each agent to interact with the environment and other agents, simulating real-world reactive behaviors. Third, `ProSim` allows *efficient* closed-loop training. During rollout, `ProSim` only runs the heavy `Encoder` and `Generator` once, while the `Policy` runs for all agents in parallel at each step. This design enables us to train `ProSim` closed-loop with full rollout of all the agents and get a complete training signal through all simulation steps.

## 4.1  Encoder

The Encoder translates the initial scene $\sigma = (M, A)$ into a set of scene tokens $F = [F_m, F_a]$, where $F_m = \{f_{m[1]}, ..., f_{m[S]}\}$ is the set of $S$ map features and $F_a = \{f_{a[1]}, ..., f_{a[N]}\}$ is the set of $N$ agent features. To enable closed-loop inference for all agents in parallel, we design $F$ to be *symmetric* across all agents and directly usable by the Generator and Policy modules for *any* agent.

We follow Shi et al. [16] and use a symmetric Encoder design. We normalize the map $M$ and actors $A$ to their local coordinate systems, and use global positions to reason about positional relationships between features with a *position-aware* attention module. The output features are then both *symmetric* and *position-aware*.

**Symmetric Feature Encoding.** Each map element $v_i$ is a set of polyline vertices representing a lane segment. These points are originally in a global coordinate system. To transform these points into an element-centered local coordinate system, we compute the geometric center and tangent direction of $v_i$ in global coordinates as $p_{m[i]}$ and $h_{m[i]}$ and transform all points in $v_i$ to the local coordinate system with $\Gamma(v_i, p_{m[i]}, h_{m[i]})$, where $\Gamma$ is the coordinate transform function. Similarly, for each historical agent state $a_{h[i]}$ in $A$, we obtain its last-step position $p_{a[i]}$ and heading $h_{a[i]}$ in the global frame. We then transform all historical states in $a_{h[i]}$ to this local coordinate system with $\Gamma(a_{h[i]}, p_{a[i]}, h_{a[i]})$. These scene features are encoded with a PointNet-like encoder [17]:[2] $f_{m[i]}^0 = \phi(\text{MLP}(\Gamma(v_i))), f_{a[i]}^0 = \phi(\text{MLP}(\Gamma(a_{h[i]})))$, where $\phi$ denotes max-pooling and $f_{m[i]}^0, f_{a[i]}^0 \in \mathbb{R}^D$ are the agent-symmetric input tokens for $v_i$ and $a_i$, respectively, with feature dimension $D$. All map and agent input tokens are concatenated together to obtain input scene tokens $F_{ma}^0$.

**Position-aware Attention.** Since each token in $F_{ma}^0$ is normalized to its local coordinate system, vanilla MHSA cannot infer the relative positional relationship between tokens. Instead, we explicitly model the relative positions between tokens with a position-aware attention mechanism. The resulting tokens we denote as $F$. Please refer to the Appendix A.1 for additional details.

## 4.2  Generator

The policy token Generator takes scene tokens $F$ and user prompts $\rho$ and outputs policy tokens for all the agents $\{\pi_1, ..., \pi_N\}$. $\rho$ contains multiple layers of information: agent static properties $a_{s[i]}$, multimodal agent-centric prompts $\rho_i$ and a scene-level text prompt $L$. Therefore, we generate the policy token $\pi_i$ with three steps. We first obtain agent policy query $q_{a[i]}$ with $a_{s[i]}$ and $F$. We then condition $q_{a[i]}$ with agent-centric prompts $\rho_i$ to obtain prompt-conditioned policy query $q_{\rho[i]}$. Finally, we embed $q_{\rho[i]}$ and $L$ together into text space and query an LLM to output the final policy token $\pi_i$. We explain these steps in details below.

**Policy Query from Scenes**. We use an MLP to encode the agent static properties $a_{s[i]}$ into an agent policy query $q_i \in \mathbb{R}^D$. To model agent-agent and agent-scene interactions, we use $Q = \{q_1, ..., q_N\}$ and scene tokens $F$ as inputs and pass them through multiple transformer layers. Each layer follows $Q' = \text{MHCA'}(\text{MHSA'}(Q), F))$, where MHCA', MHSA' denote position-aware attention modules as in Section 4.1. We take last-layer outputs as the policy queries $Q_u = \{q_{a[1]}, ..., q_{a[N]}\}$.

**Multimodal Agent-centric Prompting**. For each agent, its user-input prompt contains $|\rho_i|$ multimodal prompts $\{(x_1, \omega_1), ..., (x_{|\rho_i|}, \omega_{|\rho_i|})\}$, where $x$ is the prompt raw input and $\omega$ is the prompt type. We encode each prompt in $\rho_i$ into a D-dimensional feature $e_j = \text{Enc}_{\omega_j}(x_j)$, where $\text{Enc}_{\omega_j}$ is the feature encoder for prompt type $\omega_j$. We then aggregate all the prompts of the same agent to a compound prompt condition feature: $q_{\rho[i]}' = \phi'(\{\mathbf{e}_1, \mathbf{e}_2, \dots, \mathbf{e}_k\})$, where $\phi'$ can be any feature aggregation function (max pooling, average pooling or a learned self-attention module). We add this prompt condition feature into policy queries to get prompt-conditioned policy queries $q_{\rho[i]} = q_{a[i]} + q_{\rho[i]}'$.

**Text Prompting with LLMs**. For each scene, we have an optional user-input text prompt $L$ that contains multiple sentences that could describe agent behaviors, interactions, and scenario properties. To process the scene-level text prompt $L$ and condition all the agents, we use an LLM to comprehend

---

[2]We omit the $p, h$ parameters in $\Gamma$ for simplicity.

the natural language prompt and policy queries, and generate language-conditioned policy queries for each agent $q_{L[i]}$. In particular, we employ a LLaMA3-8B [18] model finetuned with LoRA [19] as backbone, with two adaptors to bridge the latent spaces of the LLM and the policy tokens. Please refer to the Figure A1 and Appendix A.2 for details. Finally, we obtain each agent's policy token $\pi_i$ by adding the prompt and text conditional policy queries together: $\pi_i = q_{\rho[i]} + q_{L[i]}$.

## 4.3  Policy

For rollout, at each timestep $t$ each agent $i$ independently gets its next $k$ states from a `Policy` network: $\mathbf{s}_{t:t+k}^i = \texttt{Policy}(\pi_i, \mathbf{s}_{1:t-1}, F)$. `Policy` first updates the dynamic tokens in $F$ with the new state observations $\mathbf{s}_{1:t-1}$, then it outputs agent actions for the next $k$ steps conditioned the policy token $\pi_i$. We describe these two steps below.

**Dynamic Observation Update**. Recall that $F = [F_m, F_a]$, where $F_m$ is the map feature token set and $F_a$ is the agent history token set. At timestep $t > 0$, $F_m$ is unchanged while the agent trajectory feature $F_a$ obtained at $t = 0$ is outdated. To update agent tokens with new observations, we encode $\mathbf{s}_{1:t-1}$ with symmetric feature encoding described in Section 4.1. Specifically, for each agent $i$ in $\mathbf{s}_{1:t-1}$, we have its history states $\mathbf{s}_{1:t-1}^i$, last-step position $p_{t-1}^i$ and heading $h_{t-1}^i$ in the global coordinate system. We encode it with: $f_{1:t-1}^i = \phi(\text{MLP}(\Gamma(\mathbf{s}_{1:t-1}^i, p_{t-1}^i, h_{t-1}^i)))$, where $\Gamma$ and $\phi$ are the coordinate transform and feature pooling functions defined in Section 4.1. With this operation we obtain the new agent token set $F_{a,t} = \{f_{1:t-1}^1, ..., f_{1:t-1}^N\}$ with symmetric features.

**Prompt-conditioned Action Prediction**. For each agent policy token $\pi_i$, we let it attend to all the map and agent history tokens through multiple cross-attention layers. At each layer we have $\pi_i' = \texttt{MHCA}'([\pi_i], \{F_m, F_{a,t}\})$, where MHCA is the position-aware multi-head cross attention module mentioned in Section 4.1. We extract the last-layer output $\pi_{i,t}^*$ as the action prediction feature for agent $i$ at step $t$. Then, we use an MLP to predict the state changes for the next $k$ steps $\{\Delta s_{t'}^i\}_{t'=t}^{t+k} = \text{MLP}(\pi_{i,t}^*)$. Finally, for each timestep $t' \in [t, t+k]$, we simply add the state changes to the previous step state: $s_{t'}^i = \Delta s_{t'}^i + s_{t'-1}^i$. This leads to the next $k$-step agent states $\mathbf{s}_{t:t+k}^i = \{s_t^i, ..., s_{t+k}^i\}$. We run this process for all the agents *independently* in *parallel*, supporting efficient scenario rollout.

## 4.4  Training

We train `ProSim` in a closed-loop manner. In this section we first describe the closed-loop rollout process, and then introduce the training objectives of `ProSim`. We further discuss the training details of the `Generator` LLM in Appendix A.3.

**Closed-loop Rollout.** Training a reactive traffic simulator requires strong signals of 1) how an agent's current action influences its future; 2) how each agent's action influences other agent's actions in real time. This leads us to training `ProSim` in a closed-loop manner. Specifically, given input scene and prompt $(\sigma, \rho)$, we run `Encoder` and `Generator` *once* to get the policy tokens for all the agents. Then, at each timestep, we sample new states from `Policy` for all the agents independently. We update all the agent's states with states generated by `Policy` and feed them to `Policy` in the next iteration as new observations. Finally, when $t = T$ we obtain the full rollout $\hat{\tau}$ and compute the loss with the full trajectories of all agents.

The fully differentiable design of `ProSim` makes this theoretically possible as we can directly optimize the loss with back-propagation through time in simulation. We also design `ProSim` to make it practically feasible by only running the heavy `Encoder` and `Generator` *once* in the beginning, while running the light-weight `Policy` autoregressively for all agents in *parallel* through time.

**Learning Objective.** We supervise the output $\hat{\tau} = \texttt{ProSim}(\sigma, \rho)$ against GT scenario $\tau$ with imitation learning. Specifically, an output contains $N$ agent trajectories through $T$ steps $\hat{\tau} = \{\hat{s}_1^1, ..., \hat{s}_T^N\}$. We compute the imitation learning loss with $\mathcal{L}_{\text{IL}} = \frac{1}{N \cdot T} \sum_{i=1}^N \sum_{t=1}^T d(\hat{s}_i^t, s_i^t)$, where $d$ is a distance function (e.g., Huber) and $s_i^t$ is the GT state from $\tau$. In addition to $\mathcal{L}_{\text{IL}}$, we also add a collision loss $\mathcal{L}_{\text{coll}}$ and offroad loss $\mathcal{L}_{\text{off}}$ to encourage agents to avoid collisions and stay on drivable areas. For $\mathcal{L}_{\text{coll}}$, we compute the intersection area between each pair of agents' bounding boxes across time

according to their sizes and trajectories in $\hat{\tau}$. For $\mathcal{L}_{\text{off}}$ we compute the intersection area between each agent's bounding boxes and the non-drivable area[3]. We denote the weight of $\mathcal{L}_{\text{coll}}$ and $\mathcal{L}_{\text{off}}$ as $\lambda_1$ and $\lambda_2$. Formally, the overall learning loss of ProSim is $\mathcal{L} = \mathcal{L}_{\text{IL}} + \lambda_1 \mathcal{L}_{\text{coll}} + \lambda_2 \mathcal{L}_{\text{off}}$.

## 5 ProSim-Instruct-520k

To provide realistic and diverse agent motion data with multimodal prompts for promptable closed-loop traffic simulation, we propose ProSim-Instruct-520k: a high-quality paired prompt-scenario dataset with 520K real-world driving scenarios from the Waymo Open Motion Dataset (WOMD) [20]. It includes multimodal prompts for more than 10M unique agents, representing over 575 hours of driving data. For each real-world scenario, we label a comprehensive and diverse set of prompts (action tags, text descriptions, etc.) for all agents. We ensure that these descriptions accurately correspond to agent motions in the scenario with carefully-designed labeling tools and meticulous quality assurance by human labelers. Further, the labeling tools we developed can be directly transferred to other driving data, making it easy to expand the scale of our dataset. In the remainder of this section, we describe our labeling process and metrics to evaluate promptable closed-loop traffic simulation. Details about quality assurance can be found in Appendix B.5.

### 5.1 Multimodal Prompt Labeling

Given a GT scenario $(\sigma, \tau)$ containing $N$ agents, we aim to obtain a set of multi-modal prompts for each agent $\{\rho_1, ..., \rho_N\}$ as well as a natural language prompt $L$ for the whole scenario. For each type of prompt, we carefully design a labeling tool, explained below with further details in Appendix B.

**Goal Point** is a 3-dimensional prompt. It allows the user to prompt an agent to reach a target location with a single click on the map. For an agent in the scenario, its goal point $(x, y, t)$ represents a 2D location $(x, y)$ to which this agent intends to reach at time $t$. We extract Goal Point from GT rollouts.

**Route Sketch** is a point-set prompt. It allows the user to draw a sketch on the map with the mouse as a route that they instruct an agent to follow. Route Sketches enable the users to have fine yet convenient control of the agents' routes. We randomly subsample and add Gaussian noise to the GT trajectories and view them as Route Sketch.

**Action Tag** is a categorical prompt. It allows the user to give high-level semantic behavior instruction to an agent by choosing from a set of action tags. Each action tag consists of an action type and its temporal duration $[t_s, t_e]$. We define 8 different action types: Speed (Accelerate, Decelerate, KeepSpeed, Stopping, Parked) and Turning (LeftTurn, RightTurn, Straight). For each action type, we carefully design an automatic labeling function that outputs the duration of an agent satisfying this action in the scene given GT trajectories. After obtaining tags from all action types, we conduct a postprocessing step to smooth temporal labels and remove conflicts.

**Text** is a natural language prompt. It allows the user to describe multiple agents' behaviors and the scenario properties with free-form natural language. To ensure that we have diverse language expressions while keeping the motion descriptions accurate, we prompt the Llama3-70B [18] model to generate diverse sentences given structured agent behavior descriptions (Action Tags). Please refer to Appendix B.3 for details. In total, the text prompts contain around 10M sentences.

### 5.2 Metrics

We measure the performance of promptable closed-loop traffic simulation via *realism* and *controllability*. Here we implement metrics for them. Given GT data $(\tau, \sigma, \rho)$, we evaluate a model that outputs a rollout given initial scene and prompts: $\hat{\tau} = \text{Model}(\sigma, \rho)$. We present a formal formulation for the metrics in Appendix B.4.

---

[3]We refer readers to Appendix A.3 for detailed formulations of $\mathcal{L}_{\text{coll}}$ and $\mathcal{L}_{\text{off}}$.

| Prompt | Goal Point | Route Sketch | Action Tag | Text | All w/o Text | All Types |
|---|---|---|---|---|---|---|
| ADE (m) ↓ | 0.3882 | 0.5137 | 0.6718 | 0.6937 | 0.3635 | 0.2877 |
| % Gain ↑ | 59.12% | 45.91% | 29.26% | 26.96% | 61.72 % | 69.71% |

Table 1: Realism and controllability evaluation of `ProSim`.

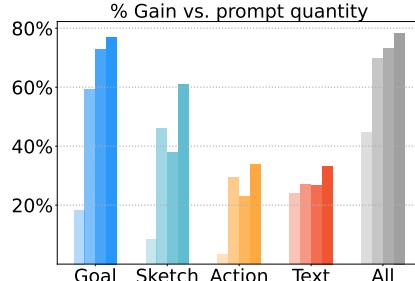

Figure 2: Prompt quantity analysis.

| Method | Composite | Kinematic | Interactive | Map-based |
|---|---|---|---|---|
| ConstVel | 0.399 | 0.225 | 0.433 | 0.453 |
| TrafficBots[5] | 0.699 | 0.430 | 0.711 | 0.836 |
| MTR+++[21] | 0.700 | 0.329 | 0.713 | 0.692 |
| VPD-Prior | 0.708 | 0.448 | 0.728 | 0.831 |
| MTR-E[21] | 0.716 | 0.426 | 0.747 | 0.841 |
| VBD[22] | 0.720 | 0.417 | 0.782 | 0.814 |
| Trajeglish[7] | 0.735 | 0.479 | 0.783 | 0.820 |
| SMART-large[8] | 0.756 | 0.477 | 0.799 | 0.862 |
| `ProSim` | 0.718 | 0.401 | 0.778 | 0.822 |

Table 2: WOMD Sim Agents Challenge 2024

**Realism**. Following prior works [2], we quantify the *realism* of the model by measuring the Average Displacement Error (**ADE**) between $\hat{\tau}$ and GT rollout: $ADE(\hat{\tau}, \tau)$, which is the mean L2 distance between agent positions and their GT across all timesteps. Lower ADE indicates better realism.

**Controllability**. We quantify *controllability* by comparing the relative improvement (**% Gain**) in realism of the model's output with and without prompts. We compute % Gain by comparing rollout ADE with and without prompt conditioning.

# 6 Experiments

## 6.1 Implementation Details

**Dataset**. We train `ProSim` on the training split of `ProSim-Instruct-520k`, which contains around 480K paired prompt-scenario data. Each scenario contains at most 128 agents, including 1.1s history and 8s future with FPS = 10 ($T = 80$). We use $k = 10$ as the steps of each policy action segment length. We use the remaining 48K scenes in `ProSim-Instruct-520k` as the testing set.

**Training**. We train `ProSim` in two stages. In the first stage, we train `ProSim` without any prompt for 10 epochs with a batch size of 64. We use the AdamW optimizer with an initial learning rate of 1e-3 and a CosineAnnealing scheduler with a 2500-step warm start. In the second stage, we fine-tune `ProSim` using all types of prompts. For each scenario, to ensure `ProSim` retains unconditional rollout capacity, we randomly mask out 50% of the prompts. In this stage, we fine-tune full `ProSim` for 5 epochs with a batch size of 16 and a learning rate of 1e-4. For LLM, we fine-tune Llama3-8B [18] using a LoRA module with $R = 16, \alpha = 0.1$. For the loss function, we set $\lambda_1 = 50.0, \lambda_2 = 5.0$.

## 6.2 Promptable Closed-loop Traffic Simulation

**Realism and Controllability.** Table 1 evaluates the realism and controllability of `ProSim` when different types of prompts are given separately, and when all types of prompts are given together, during evaluation. To simulate real-world user inputs, for each scenario, we provide 50% of all possible prompts as model input.

As can be seen, `ProSim` achieves high realism and controllability under all types of prompt. For example, given Goal Point as input, `ProSim` achieves an ADE of only 0.39m over an 8-second trajectory, which is 59.12% better than without prompt conditioning. Even the most high-level and abstract Text prompt leads `ProSim` to obtain a 26.96% Gain over the unconditional rollout. These results indicate that `ProSim` creates realistic traffic rollouts while closely following user prompts. Further, passing all prompt types as input achieves the best result of 0.28m ADE (69.7% better than without prompts), showing that `ProSim` efficiently combines and follows complex sets of multimodal prompts.

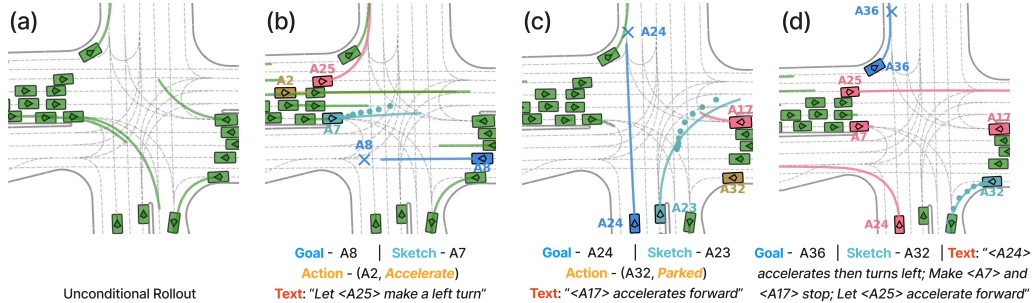

Figure 3: `ProSim` can condition on a variety of prompt types. All agents are controlled by `ProSim`.

**Qualitative Results.** We visualize the prompting capabilities of `ProSim` in two scenarios in Figures 1 and 3. Green agents are controlled by `ProSim` without prompts, while agents with other colors are controlled by the corresponding type of prompt (e.g., blue for Goal Point). The Goal Point is shown with crosses, Route Sketch with dots, while Action Tag and Text are written under figures.

Visible in Figures 1 and 3 are `ProSim`'s two core properties: It is *promptable* and *closed-loop*. Figure 3 (d) in particular shows 4 agents being prompted by a single sentence with complex high-level information, with other 2 agents simultaneously prompted by low-level goal points. Figure 3 also demonstrates how agents react to each other in *closed loop*. As can be seen, the unconditioned right-turning agent (A7) in (a) changes its behavior to yield to A24 in (c). Also, the two agents following A7 brake as A7 yields in (c). Please refer to our video for animated demos.

**Prompt quantity analysis**. Table 1 shows results of `ProSim` when we sample 50% of the prompts for evaluation. We ablate this sampling ratio in Figure 2 by setting the sampling ratio to [25%, 50%, 75%, 100%]. We show % Gain of each sampling ratio from left to right for each prompt type. We can see for most prompt types there is a clear trend that more prompts lead to higher realism gains. These results indicate that `ProSim` adapts to varying number of prompts, and the behavior generation realism scales with the number of prompts.

### 6.3 Unconditional Rollout Evaluation

As `ProSim` can run without any prompts, we can compare its performance on unconditional traffic simulation on the Waymo Sim Agent Challange [1]. In Table 2, we show the result of `ProSim` on the 2024 challenge. We compare with the SOTA method and several representative methods on the leaderboard. Overall, `ProSim` shows competitive performance. In particular, `ProSim` has a high Interactive score, showing the effectiveness of `ProSim`'s closed-loop training. On the other hand, `ProSim` has a relatively low Kinematic score (measures the diversity of agents' speed and acceleration), indicating that we can further improve `ProSim`'s output diversity during unconditional rollouts. As the focus of this work is to make `ProSim` support *promptable* and *closed-loop* traffic simulation, we leave output diversity as future research.

## 7 Conclusion

We present `ProSim`, a multimodal promptable closed-loop traffic simulation framework. Given complex sets of multimodal prompts from the users, `ProSim` simulates a traffic scenario in a closed-loop manner while instructing agents to follow the prompts. `ProSim` shows high realism and controllability given different complex user prompts. We also developed `ProSim-Instruct-520k`, the first multimodal prompt-scenario paired driving dataset with over 520K scenarios and 10M+ prompts. We believe the `ProSim` model and dataset suite will open gates for future research on promptable human behavior simulation, within and beyond driving scenarios.

**Limitations.** `ProSim` does not yet support arbitrary prompts. Complex agent interactions (e.g., *"<A0> overtakes <A1> from the left lane"*) or more complex modalities (e.g., prompt <A0> with its front-view image) are left as future work.

**Acknowledgments**

We thank Yue Zhao, Vincent Cho, Jeffrey Ouyang-Zhang and Brady Zhou for their insightful discussions.

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

# Appendix

In the appendix, we provide implementation and experiment details of our method `ProSim`, our dataset `ProSim-Instruct-520k` as well as additional experiment results. In Section A, we present details of `Encoder` and `Generator` as well as the training process. In Section B, we present details of labeling, metric and quality assurance processes of our dataset `ProSim-Instruct-520k`. Finally, in Section C, we show additional quantitative and qualitative results of `ProSim`.

## A  `ProSim`

### A.1  `Encoder`: Position-aware Attention Details

To model relationships between scene tokens and aggregate features, we pass $F_{ma}^0$ through multiple Transformer layers. In most existing methods, each layer follows $F_{ma}^l = \text{MHSA}(F_{ma}^{l-1})$, where $\text{MHSA}$ denotes multi-head self-attention and $l$ is the layer index. However, since each token in $F_{ma}^0$ is normalized to its local coordinate system, basic $\text{MHSA}$ cannot infer the relative positional relationship between tokens. Instead, we explicitly model the relative positions between tokens with a position-aware attention mechanism. For scene token $i$, we compute its relative positional relationship with scene token $j$ with:

$$p_{ma[i,j]} = \text{Rot}(p_{ma[j]} - p_{ma[i]}, -h_{ma[i]}), \quad h_{ma[i,j]} = h_{ma[j]} - h_{ma[i]}, \tag{2}$$

where $p_{ma[i,j]} \in \mathbb{R}^2$ and $h_{ma[i,j]} \in \mathbb{R}$ are the relative position and heading of token $j$ in token $i$'s coordinate system, $\text{Rot}(\cdot)$ is the vector rotation function. We denote this paired relative position as $r_{ma[i,j]} = [p_{ma[i,j]}, h_{ma[i,j]}]$. Then, we perform position-aware attention for token $i$ with:

$$\begin{aligned} f_{ma[i]}^l = \text{MHSA}(\text{Q} &: [f_{ma[i]}^{l-1}, \text{PE}(r_{ma[i,i]})], \\ \text{K} &: \{[f_{ma[j]}^{l-1}, \text{PE}(r_{ma[i,j]})]\}_{j \in \Omega(i)}, \\ \text{V} &: \{[f_{ma[j]}^{l-1} + \text{PE}(r_{ma[i,j]})]\}_{j \in \Omega(i)}), \end{aligned} \tag{3}$$

where PE denotes positional encoding and $\Omega(i)$ is the scene token index of the neighboring tokens of $i$. In our experiments, we set $\Omega(i)$ to contain the nearest 32 tokens of $i$ according to their positions. Note that the above result remains the same regardless of which global coordinate system we use for the scene input $\sigma = (M, A)$. With this formulation, we model the relative position relationship between different scene tokens symmetrically. At each layer, we apply Equation 3 to all scene tokens in parallel. We denote this position-aware attention module as:

$$F_{ma}^l = \text{MHSA}'(F_{ma}^{l-1}, P_{ma}, H_{ma}) \tag{4}$$

Note that this position-aware modification can be similarly applied to multi-headed cross-attention `MHCA'`, which we will use later in the `Generator` and `Policy` modules. Finally, we obtain the last-layer token features as scene tokens $F = [F_m, F_a]$.

### A.2  `Generator`: Language Prompting Details

For each scene, we have an optional user-input text prompt $L$ that contains multiple sentences that could describe agent behaviors, interactions and scenario properties. To make it easy to refer to different agents in the scene, we ask the user to use a specific format "<A[i]>" when mentioning the $i$-th agent. For example, to instruct agent $a_1$ to stop, the user would say *"let <A1> stop"*.

To process the scene-level text prompt $L$ and condition all the agents, we use an LLM to comprehend the natural language prompt and policy features, and generate language-conditioned policy features for all agents. To do this, we use a LLaMA3-8B model finetuned with LoRA as backbone, as well as two adaptors to bridge the latent spaces of LLM and policy tokens. We show an overview of our model in Figure A1.

Specifically, we use a two-layer MLP adaptor to convert all the policy features to in the LLM's feature space $\{t_{a[1]}, ..., t_{a[N]}\} = \text{MLP}(\{q_{\rho[1]}, ..., q_{\rho[N]}\})$. This operation enables the LLM to take

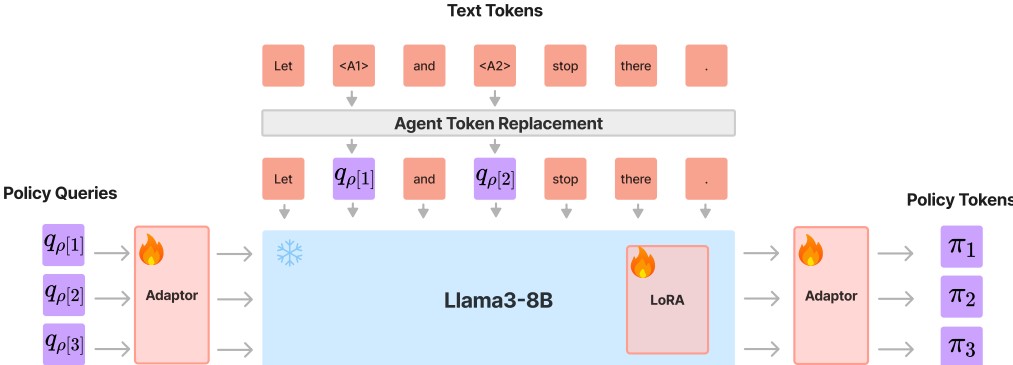

Figure A1: Language condition encoder for `Generator`.

and comprehend agent policy features in the text space. Next, we use LLM's text tokenizer and input embedding to obtain text tokens from the natural language prompt $L$ $\{t_{L[1]}, ..., t_{L[O]}\}$, where $O$ is the number of text tokens in $L$.

Note that some tokens of $L$ specifically mentions indexed agents (e.g., "<A1>"). To help LLM understanding the correspondence between agent reference and their policy tokens, we directly replace these reference text tokens with agent policy tokens. Specifically, for an agent-reference text token $t_{L[j]}$, we replace it with the corresponding agent policy token $t_{L[j]} \leftarrow t_{a[i]}$ given $t_{L[j]}$ corresponds to "<A[i]>" To make sure each "<A[i]>" is tokenized to a single token, we add them as new tokens to the text tokenizer. We call this step "Agent Token Replacement" in Figure A1.

After getting all the text and agent token features, we concatenate them to create the complete input sequence $\{t_{L[1]}, ..., t_{L[O]}, t_{a[1]}, ..., t_{a[N]}\}$ and feed it to the LLM. We then extract the hidden features for the last $N$ tokens from the last LLM layer $\{t'_{a[1]}, ..., t'_{a[N]}\}$, which contains text-contextualized policy features for each agent. Next, we use an MLP adaptor to convert these features back to get the text-conditional policy features $\{q_{L[1]}, ..., q_{L[N]}\} = \text{MLP}(\{t'_{a[1]}, ..., t'_{a[N]}\})$.

Finally, for each agent, we obtain its policy token $\pi_i$ by adding the prompt and text conditional policy tokens together, $\pi_i = q_{\rho[i]} + q_{L[i]}$.

## A.3 Training

**LLM Pretraining.** As described above, we train the LLM in `Generator` with LoRA to comprehend and generate policy token features. However, we found that directly training the LLM with the closed-loop imitation loss leads to inferior text-prompting performance. We conjecture this is because at the early training stage the LLM is not prepared to interact with the policy tokens. Meanwhile, the rollout loss can be decently optimized without using the LLM's output, giving little signal for LLM to learn.

To deal with this issue, we propose to pretrain the LLM to have the capacity to interact with policy token features. To this end, we first train a `ProSim` model without using text prompts and the LLM with the rollout loss $\mathcal{L}$. Next, we add a simple MLP layer after the `Generator` to predict the goal point of each agent given its policy token $\pi$. We supervise this task with an MSE loss $\mathcal{L}_{\text{goal}}$ using GT goal points. This task is much simpler than the full task while enforces the LLM to interact with policy tokens. To pretrain the LLM, we fix all other modules and only train the LLM and this new MLP with $\mathcal{L}_{\text{goal}}$. Here we only use text as the prompt in inputs. After pretraining, the LLM learns to predict the goal intention of each agent from text prompt and add that information to $\pi$, making it already useful for the `Policy`. Finally, we discard the goal-prediction MLP and train the full `ProSim` module with all types of prompts and the complete loss $\mathcal{L}$. In our experiments, we pretrain the LLM on `ProSim-Instruct-520k` for 5 epoches before using it in the full `Generator` module.

**Collision loss.** For collision loss $\mathcal{L}_{\text{coll}}$, we aim to compute the overlapping area between each pair of agent using their bounding boxes through the full rollout trajectory. To this end, at each timestep, for each agent we compute their occupation polygon using their size, position and heading at this timestep. Then, we measure the overlapping area of each agent polygon pair by computing the signed distance between these polygons, where positive distance indicate no collision while negative distance collision indicate a collision. Specifically, we compute the signed distance between polygons A and B with the distance between the origin point and the Minkowski sum A + (-B). For all the signed distance, we compute the loss value by first setting the positive distance to 0 (no collision), and then taking the negative of the rest of signed distances to penalize collisions. We compute the average over all agents through all timesteps to obtain the final $\mathcal{L}_{\text{coll}}$. We implement this loss function by referring to the Waymo Open Sim Agent Challenge collision metric.

**Offroad loss.** For offroad loss $\mathcal{L}_{\text{off}}$, we aim to compute the overlapping area of each agent and the offroad areas through the full rollout trajectory. To this end, we use the similar strategy as in the collision loss. Specifically, at each timestep we obtain all the agent polygons in the same way as in the collision loss. Then, we compute the signed distance between the four corners of each polygones to a densely sampled set of road edges. Similarly, positive distance indicate no offroad while negative distance indicate being offroad. Finally, We average over all negative distances for all agents through all timesteps to obtain the final offroad loss. We implement this loss function by referring to the Waymo Open Sim Agent Challenge offroad metric.

# B  `ProSim-Instruct-520k`

## B.1  Route Sketch Labeling Details

Compared with the real trajectory, points in the route sketch are sparse, noisy, and incomplete. We simulate these effects with the following steps. First, we extract each agent's complete trajectory from $\tau$. Then, we process this point set by 1) uniformly subsampling the points for sparsity; 2) adding random noise to each point; 3) randomly sampling a consecutive subset. For each agent, its route sketch is a set of ordered 2D points representing sketch points on the map. The number of points in route sketch could be different across agents. In our experiment, we use a uniform subsample rate of 5. We than add random noise with standard variation of 0.1 meter. Finally, we ensure the randomly sampled consecutive subset contains at least 5 points.

## B.2  Action Tag Labeling Details

For each action type, we carefully design a heuristic function that takes an agent's trajectory $\mathbf{s}_{t_1:t_2}$ from step $t_1$ to $t_2$, and outputs a binary label whether $\mathbf{s}_{t_1:t_2}$ satisfies the condition of this action. Then, we can run this function across the full rollout with sliding window and temporal aggregation to obtain the full duration $[t_s, t_e]$ that this motion tag is valid. For each agent, we run all the motion tags with this method we obtain a set of motion tags. Finally, we post-process the labels to remove conflicting motion tags and temporal noises. Please refer to our codebase for the heuristic function implementation details upon release.

## B.3  Natural Language Labeling Details

For each scenario, we provide the LLM model with agent properties (name and type) as well as all of their agent tags (action type and duration). We then prompt LLM to output 20 different sentences, each describing the agent behavior or scenario properties in natural language. To obtain interesting and diverse language description of the scenario, in the system prompt we instruct the LLM to 1) describe temporal transition of agent behavior (e.g., "Let <A1> change to the left lane and then make a left turn."); 2) describe scenario properties (e.g., "This is a busy scene with most agents accelerating"); 3) describe relationships of different agents (e.g., "Let <A1>, <A2>, <A3> keep their own lanes simultaneously"). We concatenate these sentences together to form the prompt $L$ for each scenario.

Here we show full prompt we used for the LLM labeling:

Prompt 1: Full prompt for LLM labeling

```
Example input:
  Vehicle Agents:
      ['<ego>', '<71f1c>', '<df6a1>', '<dad99>']
  Pedestrian Agents:
      ['<a261a>', '<191e8>']
  Motorcycle Agents:
      ['<d3ddc>', '<8cc93>', '<73c13>', '<d6a9e>']

  Agent to Agent:
    ParallelDriving - Agent (Left):<d6a9e>, Agent (Right):<dad99>, Start:50, End:80
    ByPassingRight - Agent (Right, Faster, Overtaking):<ego>, Agent (Left, Slower, Overtaken
      ):<dad99>, Start:30, End:65

  Agent Behavior:
    Decelerate - Agent:<d3ddc>, Start:0, End:5
    Decelerate - Agent:<ego>, Start:30, End:55
    Decelerate - Agent:<dad99>, Start:35, End:45
    Decelerate - Agent:<d6a9e>, Start:45, End:80
    KeepLane - Agent:<d3ddc>, Start:0, End:60
    KeepLane - Agent:<dad99>, Start:30, End:80
    KeepLane - Agent:<d6a9e>, Start:40, End:80
    KeepSpeed - Agent:<191e8>, Start:0, End:20
    KeepSpeed - Agent:<ego>, Start:0, End:30
    KeepSpeed - Agent:<d6a9e>, Start:40, End:45
    Parked - Agent:<a261a>, Start:0, End:25
    Parked - Agent:<df6a1>, Start:0, End:80
    RightLaneChange - Agent:<ego>, Start:0, End:15
    Straight - Agent:<191e8>, Start:0, End:20
    Straight - Agent:<d3ddc>, Start:0, End:70
    Straight - Agent:<ego>, Start:0, End:80
    Straight - Agent:<dad99>, Start:30, End:80
    Straight - Agent:<d6a9e>, Start:40, End:80
    Stopping - Agent:<d3ddc>, Start:5, End:70
    Stopping - Agent:<dad99>, Start:45, End:80
    LeftLaneChange - Agent:<ego>, Start:15, End:65

Example output:

Here are the 20 commands for the simulation:
  "<ego> bypasses <dad99> from the right lane side when <dad99> is driving slower and
    finally stopping."
  "Do right lane change <ego> at the start of the simulation."
  "Motorcycle <d3ddc> decelerates early on."
  "After finishing the initial maneuver, redirect <ego> to occupy the left lane."
  "Make <dad99> and <d6a9e> to drive parallel, with <d6a9e> on the left."
  "<ego> slows down following the sequence of lane changes.."
  "All vehicles, except for <ego>, <dad99>, and <d6a9e>, remain parked."
  "Let <ego> maintain a steady speed after decelerating."
  "Command <d3ddc> cyclist to come to a complete stop after its initial slowdown."
  "Instruct <dad99> to decelerate and then stop towards the end of the scenario."
  "Keep <d6a9e> in its lane after it finishes driving parallel."
  "Walking person <191e8> should keep a steady pace before stopping."
  "<d3ddc> resumes moving straight after stopping."
  "Keep all parked vehicles stationary to represent a low-activity scene."
  "Direct <dad99> car to travel straight for an extended period after decelerating."
  "Ensure <ego> car moves straight throughout the simulation."
  "After slowing, instruct <d6a9e> to continue on a direct trajectory."
  "Let the pedestrian <a261a> standstill in the scene."
  "Emphasize the limited activity within the scene, highlighting agents either stopping or
    staying within their lanes."
  "Pedestrian <191e8> walks straight."
```

We also show two LLM output examples for two scenarios:

Prompt 2: Text prompt labeling example 1

```
Here are the 20 commands for the simulation:

1. "Have <43> accelerate and make a left turn throughout the entire simulation."
2. "Initially, <58> accelerates, but then slows down and comes to a stop."
3. "Keep <10> moving straight while it decelerates early on."
4. "Make <48> decelerate and then turn left towards the middle of the simulation."
5. "Command <39> to accelerate and drive straight after 40 seconds."
6. "Make <7> accelerate rapidly towards the end of the simulation."
7. "<ego> remains parked and stationary throughout the entire scenario."
8. "Direct <11> to stay parked for the entire duration of the simulation."
9. "Most vehicles, except for a few, are parked and stationary at the start of the
    simulation."
10. "After 40 seconds, <14> starts moving after being parked initially."
11. "Make <4> change direction with a right turn after being parked for a while."
12. "Let <48> decelerate and come to a stop before making a left turn."
13. "Have <58> accelerate initially, but then slow down and stop."
14. "Command <45> and <56> to stop after 40 seconds."
15. "Instruct <32> to stop after 15 seconds and then stay stationary."
16. "Make <15> stop and stay stationary throughout the entire simulation."
17. "Have <50>, <52>, and <58> accelerate and drive straight throughout the simulation."
18. "Keep <24> moving straight after 65 seconds."
19. "Make <39> accelerate and drive straight after being parked initially."
20. "Emphasize the dynamic nature of the scene, with agents accelerating, decelerating, and
    changing directions."
```

Prompt 3: Text prompt labeling example 2

```
Here are the 20 commands for the simulation:

1. "<653> accelerates while making a left turn throughout the entire simulation."
2. "Have <654> maintain a steady acceleration from start to finish."
3. "<661> gradually slows down and drives straight throughout the scenario."
4. "<666> decelerates and moves straight without any turns or stops."
5. "Keep <1465> cycling in a straight line for the entire 8 seconds."
6. "Make <651> drive straight without any changes in speed or direction."
7. "Ensure <659>, <660>, <662>, <664>, and <ego> remain parked and stationary throughout the
    simulation."
8. "After accelerating, have <653> continue moving in a straight line."
9. "Instruct <661> to decelerate and then maintain a steady speed."
10. "Make <654> overtake <661> from the left lane."
11. "<663> follows <661> at a steady pace, maintaining a safe distance."
12. "Command <653> to merge into the lane where <661> is driving."
13. "Direct <1465> to pass <651> on the right side."
14. "Have <661> change lanes to the left and then continue driving straight."
15. "Make <654> drive parallel to <653> on the right side."
16. "<ego> remains stationary, observing the surrounding traffic."
17. "After accelerating, have <653> change lanes to the right."
18. "Ensure <660> and <664> remain parked, blocking the left and right lanes respectively."
19. "Instruct <663> to decelerate and then stop behind <661>."
20. "The bicycle <1465> cycles past the parked vehicles, maintaining a steady pace."
```

## B.4   Metric Formulation

In our paper, we provide two metrics of promptable closed-loop traffic simulation to measure *realism* and *controllability*. Here we link these metrics to the problem formulation in Section 3. Recall that we formulate promptable closed-loop traffic simulation as

$$p(\mathbf{s}_{1:T}|\sigma, \rho) = \prod_{t=1}^{T} \prod_{i=1}^{N} p(s_t^i|\mathbf{s}_{1:t-1}, \sigma, \rho). \tag{5}$$

| Metric | ADE ↓ | Gain ↑ | ADE ↓ | Gain ↑ | ADE ↓ | Gain ↑ | ADE ↓ | Gain ↑ |
|---|---|---|---|---|---|---|---|---|
| Prompt | Goal + Sketch | | Goal + Text | | Sketch + Action | | Sketch + Text | |
| `ProSim` | 0.4845 | 48.98% | 0.5983 | 37.00% | 0.5588 | 41.16% | 0.5698 | 40.00% |
| Prompt | Goal + Action + Sketch | | Goal + Action + Text | | Text + Sketch + Action | | All Types | |
| `ProSim` | 0.3635 | 61.72% | 0.5663 | 40.37% | 0.5311 | 44.08% | 0.2877 | 69.71% |

Table A1: Controllability evaluation of `ProSim`

Given GT data $(\tau, \sigma, \rho)$, *realism* measures the probability of the real rollout under the model distribution $p(\tau|\sigma, \rho)$. In our main paper, we implement this metric with **ADE**$(\hat{\tau}, \tau)$.

On the other hand, *controllability* measures how well the model follows the prompt $\rho$. We quantify this by comparing the model's realism gain against the unconditional model rollout $p(\tau|\sigma, \rho) - p(\tau|\sigma)$. In our main paper, we implement this metric with relative improvement (**% Gain**) in realism of the model's output with and without prompts. We compute % Gain by comparing rollout ADE with and without prompt conditioning. : % Gain $= \frac{\text{ADE}(\bar{\tau}, \tau) - \text{ADE}(\hat{\tau}, \tau)}{\text{ADE}(\bar{\tau}, \tau)} \times 100\%$.

## B.5 Quality Assurance

To ensure the prompts we generate faithfully reflect the agent behaviors in the scenario, we conduct a careful quality assurance process with human effort. As Goal Point and Route Sketch are directly modified from real trajectories, there is no need to check their accuracy. On the other hand, the accuracy of Text is largely dependent on the accuracy of the Action Tag as it stems from Action Tags of all agents in a scene. Therefore, we focus on checking the quality of Action Tag. For each action types, we ask human labelers to manually check whether the labeled action tag is accurate both semantically and temporally by viewing the rollout videos. At each round, we ask human labelers to check 100 motion tag examples for each action type. If the qualification rate of a certain action type is below 85%, we rewrite the heuristic function of this action type according to the human feedback, relabel the motion tags of this type and ask for another round of human checking. We continue this process until all the action types pass the quality threshold.

We show the interface we developed for human labelers in Figure A2. This interface allow human labelers to go through and rewind the scenario easily with the interactive progress bar. For each scenario with multiple action tags, the interface let the labeler to go through all the action tags all together. This allows the human labeler to QA multiple motion tags of the same scenario very efficiently. In average, we found human labelers take around 10 seconds to give the QA output for each motion tag. Additionally, we ask human labelers to give a QA output for each tag, choosen from Correct, Wrong Action, Wrong Time, Wrong Agent, and Need Attention (unsure or other types). These different types of QA error tags provide us useful feedback to improve our heuristic functions.

## C Additional results

### C.1 Controllability analysis

In our paper, we show benchmark results of 5 different prompt combinations. Aside from these combinations shown in the paper, we also show the % Gain results of other prompt combinations in Table A1. We can see from the Table A1 that `ProSim` achieves consistent gain with different kinds of prompt modality combinations. These results show that `ProSim` allows users to freely combine different prompt modalities with high controllability.

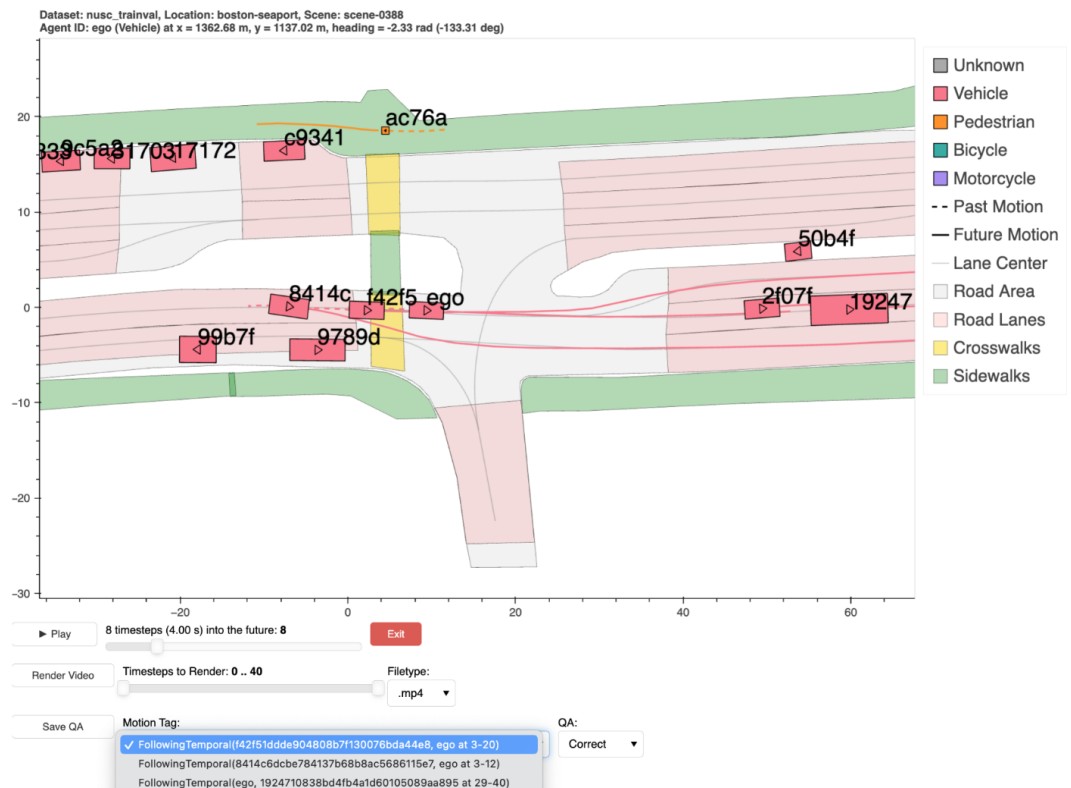

Figure A2: Interface used by human labeler for Quality Assurance.

| Sample batch size | 1 | 4 | 8 | 16 |
|---|---|---|---|---|
| Unconditional | 248.64 | 79.89 | 53.08 | 43.09 |
| All Prompting w/o Text | 266.31 | 82.65 | 54.82 | 44.02 |
| All Prompting with Text | 365.98 | 105.94 | 66.54 | 49.99 |

Table A2: Average inference time of `ProSim` in milliseconds for 1 scenario with 64 agents given different sample batch sizes.

## C.2 Runtime analysis

In this section, we provide a thorough rollout time analysis of `ProSim`. Specifically, we use a single NVIDIA A100 GPU to measure the runtime of `ProSim` under three different settings: 1) sample batch size; 2) number of simulated agents per scenario and 3) number of prompts used per scenario. In each experiment, `ProSim` is tasked with generating 8-second long trajectories for up to 128 simulated agents per scenario.

| # Simulated agents | 16 | 32 | 64 | 128 |
|---|---|---|---|---|
| Unconditional | 239.51 | 240.76 | 247.69 | 267.06 |
| All Prompting w/o Text | 249.31 | 251.56 | 262.54 | 283.54 |
| All Prompting with Text | 324.71 | 332.43 | 353.52 | 394.34 |

Table A3: Average inference time of `ProSim` in milliseconds for 1 scenario with batch size 1 given different numbers of simulated agents.

| Quantity of prompts | 25% | 50% | 75% | 100% |
|---|---|---|---|---|
| All Prompting w/o Text | 258.01 | 258.86 | 260.67 | 259.96 |
| Text Prompting Only | 327.53 | 341.97 | 346.39 | 352.99 |
| All Prompting with Text | 337.94 | 355.03 | 363.55 | 372.58 |

Table A4: Average inference time of `ProSim` in milliseconds for 1 scenario with batch size 1 and 64 agents given different prompt quantities.

Firstly, in Table A2 we conduct ablation study on inference batch size (number of scenarios per batch). we prompt 50% of all agents at random (also randomly selecting a prompt type, i.e., goal, sketch, action, text). This table shows that `ProSim` is able to achieve very high inference efficiency, especially when we use batched scenario inference. Notably, when sample batch size is 16, `ProSim` can simulate an 8-second scenario with 64 agents and all kinds of prompts (including text) within 50ms on average. This table shows that `ProSim` can be highly efficient with parallel computation.

Next, in Table A3 we do ablation study on the number of simulated agents per sceanrio, where we set batch size equal to 1. This table shows `ProSim` is able to easily scale to a large number of agents (up to 128) with minimal increases in inference time. Specifically, inference time only grows by 12% from 16 agents to 128 agents. Finally, in Table A4, we vary the number of prompts used per scenario. This table shows ProSim can maintain efficiency even with a large number of prompts.

We achieve the high efficiency of `ProSim` with careful designs. Firstly, `ProSim` only run `Encoder` and `Generator` once per scenario, regardless of the number of agents in the scenario. For `Encoder`, the main bottleneck is from the many scene tokens, so agent tokens don't affect runtime much. For the `Generator`, the agent-centric prompt encoders are very lightweight and not a problem. As for the LLM module, the main bottleneck again is from text tokens, not agent tokens. Overall, the runtime of the `Encoder` and `Generator` doesn't increase much with more agents.

On the other hand, `ProSim` runs the `Policy` for all agents in parallel. We make this possible by letting each agent act individually given only its own scene and policy tokens (Section 4.3). This design not only realistically reflects how agents interact in the real world, but also allows for batched policy inference at each timestep for all agents (even across different scenarios). For example, given $M$ scenarios, each with $N$ simulated agents, at each timestep we conduct $M \times N$ policy inferences in parallel within a single batch. This operation allows us to achieve very high inference throughput on GPUs.

### C.3 Conflicting prompting analysis

Recall that when we prompt one agent with prompts with multiple modalities, we always provide prompts with the similar semantics, as this is the expected way users will use `ProSim` for. We also train `ProSim` with this manner. However, it is interesting to see what will happen if for a single agent we provide multiple prompts with different or conflicting semantics simultaneously. This experiment could show which types of prompt `ProSim` is more sensitive to.

Therefore, in this section, we investigate `ProSim`'s behavior if multiple prompts with different semantics are given simultaneously to a single agent. We provide a case study showing how ProSim deals with prompts with different semantics in Figure A3.

In this experiment, we prompt agent A8 with four different prompts,one modality at a time, as in Figure A3 (1)-(4). Note how `ProSim` is able to make A8 follow different modalities of prompts closely. To investigate the case when we provide multiple prompts with different semantics simultaneously, in (5)-(12) we prompt agent A8 with 2-3 prompts with conflicting semantics and modalities together. For example, in (7) we prompt A8 with a Goal point to turn left and with Text to turn right. Here are our observations:

- Goal prompts dominate over other prompt types. Whenever a Goal prompt is provided, agents will try to follow it first and tend to ignore other prompt types, as can be seen from (5) and (7).

- Similar to Goal, Sketch also dominates over Action and Text, as shown in (8) and (9). Note that in (6) when Sketch and Goal appear together, the agent does not strictly follow Goal, but rather tries to interpolate between the Goal and Sketch points. This indicates that Sketch has similar controllability as Goal.

- Action and Text provide more abstract instructions and are therefore dominated by more explicit Goal and Sketch prompts. When Action and Text appear together in (10), we can see the agent also tries to interpolate between Action and Text (turning right but also stopping, a mix between (3) and (4)). This also indicates that Text has similar controllability as Action.

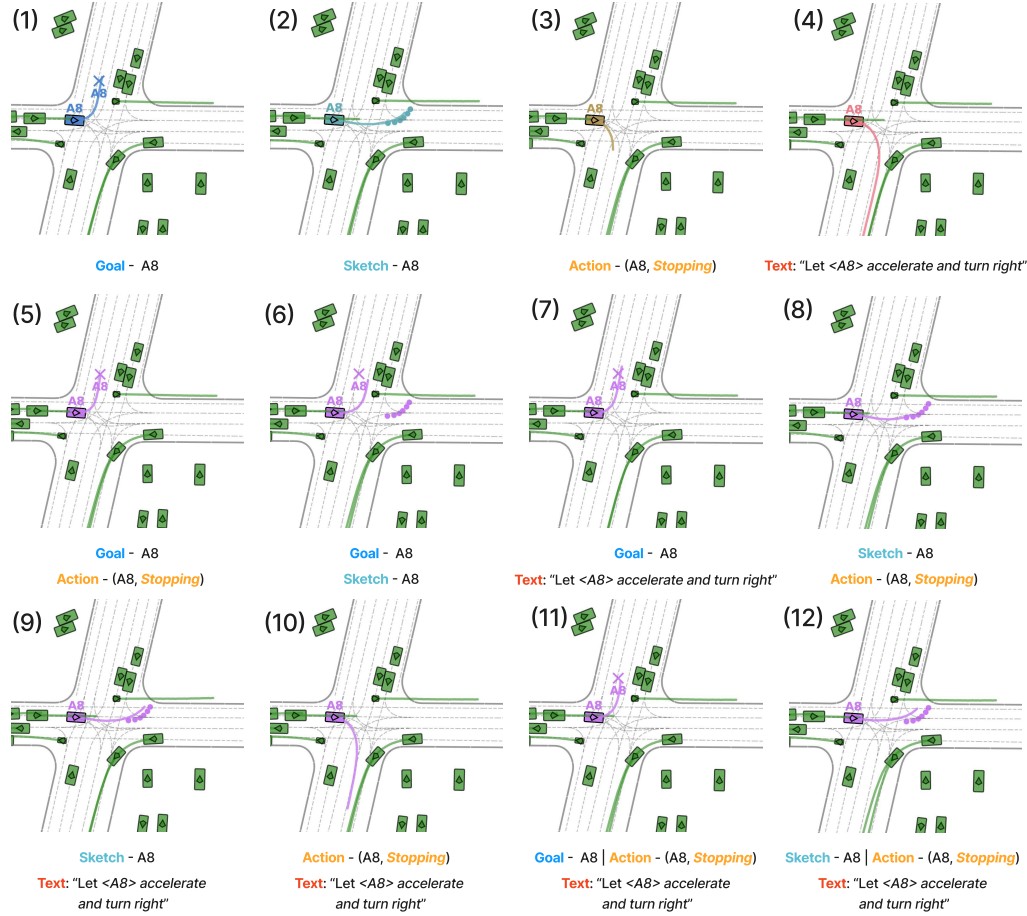

Figure A3: Prompting the same agent with prompts with different semantics. All agents are controlled by `ProSim`. Green agents are unconditional agents. On the first row, in each column agent A8 is prompted by a single prompt modality that has distinct semantics. On the last two rows, agent A8 is simultaneously prompted by more than 2 prompt modalities with conflicting semantics.

Overall, we found that, in terms of controllability, when multiple semantically-conflicting prompts appear, Goal > Sketch > Action ≃ Text. This rank resonates with the controllability scores in Table 1. We conjecture that this is because Goal and Sketch prompts contain more explicit information while Action and Text prompts are more abstract. Therefore, during training, `ProSim` learns to focus more on Goal and Sketch prompts more (which helps it reduce losses more effectively).

