# OpenReview forum: "Promptable Closed-loop Traffic Simulation"
_robot-learning.org/CoRL/2024/Conference — CoRL 2024_

### Official Review · Reviewer_w2z7 · 2024-07-18
**Nice paper**

**Originality:** 4
**Technical Quality:** 5
**Clarity Of Presentation:** 5
**Potential Impact:** 4
**Recommendation:** 3
**Confidence:** 4

**Review:**

# Clarity

The paper is well-written and easy to follow.

# Originality

The framework is novel. To my knowledge there is no such a method that can take multiple prompts to control the scene generation. The hierarchal motion generator is also very interesting.

# Strengths

1. The paper covers different forms of prompts and align them into the same framework.
2. The dataset is a good contribution to the community.
3. The paper is well-written.
4.  The idea of closed-loop training, though not novel in motion prediction, integrates well with other part of the model and is promising.


# Weaknesses

1. Table 2 does not have SOTA baselines, and it doesn't present the composite metric. This is very confusing as the golden standard of WOSAC is the composite metric.
2. The method doesn't seem to be scale as the number of agents increase.
3. I have concern on the usefulness of the text prompt. It seems that the text prompt is just the action tags (left / forward / right) in the textual format.

**Quality Of The Limitations Section:**

2

**Questions For Rebuttal:**

1. Are the LORA of LLM trained together with the whole pipeline in closed-loop manner?
2. Is it possible to use text prompt to prompt the behaviors of multiple agents? I think it's possible. Have you every tested this: as the number of agents being prompted increase, the controllability saturates.
3. In limitation you said complex agent interaction is not promptable right now. Can I understand this as the autolabeling mechanism when you creating the dataset limits the diversity of text prompt. The text prompt is actually the action tag in text format. As in Figure 3, we can see the text prompts are basically "some agents make a left turn / move forward". So the so call text prompt is not that useful as you claim.
4. As I said in Weakness 1, why the WOSAC result is not presented properly?
5. In MTR++, they use 64 queries to increase the diversity of the output. I wonder whether the "unconditional generation", and as well as the WOSAC result, can be improved if we discover say 64 "common prompts". (Note that 64 is for motion prediction, in WOSAC the required modes are 32).
6. How many agent are simulated?
7. In WOSAC they require you to simulate all 128 agents. How you address the scalability issue if you are simulating 128 agents?

**Robotics Focus:**

1

**Summary Of Paper:**

A closed-loop traffic simulation framework supporting multiple prompts, including coordinates and text. A hierarchal motion generator is used. New dataset proposed.

**Summary Of Recommendation:**

The paper is sound and well-written, with sufficient novelty and contribution to the community.

---

### Official Review · Reviewer_Vo5Q · 2024-07-20
**This paper propose a promptable closed-loop traffic simulation framework. The content is comprehensive, but the novelty is limited.**

**Originality:** 3
**Technical Quality:** 3
**Clarity Of Presentation:** 4
**Potential Impact:** 2
**Recommendation:** 2
**Confidence:** 4

**Review:**

Pros:
1. The paper introduces a novel promptable closed-loop traffic simulator.
2. The simulator supports four types of prompts, enhancing realism. Combined prompts lead to higher realism results.
3. Qualitative results demonstrate the simulator's controllability and the effectiveness of its interactivity modeling.
4. The simulator achieves competitive results in the Waymo Sim Agent Challenge.
5. The authors have built a dataset, which is claimed to be released.

Cons:
1. The controllability aspect needs further exploration. The impact of different prompts on agent behavior should be analyzed in more depth. For example, what are the effects of different prompts on agent interactions? What happens if multiple prompts with different semantics are given simultaneously? Can the simulator generalize to corner cases where agent behaviors deviate from the conventions recorded in the training dataset?
2. In Figure 3(d), agents seem to be represented with unique indexes in the text prompt. How to encoder this information in the network and bridge the text prompt and numeric features?
3. In Figure 2, the prompt effectiveness of sketch and action is not monotonically increasing, which lacks detailed analysis.
4. The diversity of the simulation results is limited.
5. The simulation efficiency should be demonstrated and compared.

**Quality Of The Limitations Section:**

2

**Questions For Rebuttal:**

Please refer to cons.

**Robotics Focus:**

3

**Summary Of Paper:**

This paper introduces ProSim, a promptable closed-loop traffic simulation framework that effectively models interactivity while maintaining realism and controllability. The paper is well-written and easy to follow. However, the prompting method can be seen as a straightforward extension of common multi-agent prediction or simulation frameworks. The improvement in ADE results with the proposed prompts, such as GT goal points, is expected. Deeper analysis and more results on long-tail scenarios are desired.

**Summary Of Recommendation:**

This paper propose a promptable closed-loop traffic simulation framework. The content is complete, but the novelty is limited.

---

### Official Review · Reviewer_M9no · 2024-07-20
**Promptable traffic simulation is an important topic for autonomous driving simulation. ProSim provides a LLM-based method. The dataset will be helpful for the sim agent community.**

**Originality:** 3
**Technical Quality:** 4
**Clarity Of Presentation:** 3
**Potential Impact:** 3
**Recommendation:** 4
**Confidence:** 4

**Review:**

1. ProSim is a first-of-its-kind promptable closed-loop traffic simulation framework. The ProSim-Instruct-520k, a large-scale multimodal prompt-scenario driving dataset, is the first driving dataset with semantic-rich agent motion labels and text captions.
2. This development including ProSim and the dataset will be helpful for the sim agent and traffic simulation community.
3. This paper is easy to follow and well written.
4. The experimental results prove the effectiveness of the proposed ProSim. ProSim achieves high prompt controllability given different user prompts, while reaching competitive performance on the Waymo Sim Agents Challenge when no prompt is given.

There are some minor questions for Prosim.
5. How to make the representation alignment (encoder) for the LLM backbone to understand the complex map and agent information without QAs?
6. I suggest that the author can provide the inference time analysis for the traffic simulation.
7. How to make a scene initialization for the Waymo Open Motion Dataset?

**Quality Of The Limitations Section:**

3

**Questions For Rebuttal:**

How to make the representation alignment (encoder) for the LLM backbone to understand the complex map and agent information without QAs?

**Robotics Focus:**

3

**Summary Of Paper:**

ProSim is a first-of-its-kind promptable closed-loop traffic simulation framework. The ProSim-Instruct-520k, a large-scale multimodal prompt-scenario driving dataset, is the first driving dataset with semantic-rich agent motion labels and text captions.

**Summary Of Recommendation:**

ProSim is a first-of-its-kind promptable closed-loop traffic simulation.

---

### Author Rebuttal · Authors · 2024-08-08

In the rebuttal file, we provide a **pdf** file for the figures of the deeper analysis of the controllability of ProSim. Please refer to our replies to reviewers for the result analysis.

---

### Decision · Program_Chairs · 2024-09-04

**Decision:**

Accept

**Comment:**

Strengths:
- Novel promptable closed-loop traffic simulation framework supporting multiple prompt types (coordinates, text, etc.)
- Creation of a large-scale multimodal prompt-scenario driving dataset (ProSim-Instruct-520k)
- Well-written paper that is easy to follow
- Competitive performance on Waymo Sim Agents Challenge, with high prompt controllability

Weaknesses:
- Limited analysis of prompt effects on agent interactions and corner cases
- Scalability concerns as number of agents increases
- Questions about usefulness/diversity of text prompts beyond basic action tags
- Lack of proper SOTA baselines and composite metric results for Waymo Open Sim Agents Challenge